# Patients with Small Acetabular Cartilage Defects Caused by Femoroacetabular Impingement Do Not Benefit from Microfracture

**DOI:** 10.3390/jcm11216283

**Published:** 2022-10-25

**Authors:** Moritz Riedl, Ingo J. Banke, Jens Goronzy, Christian Sobau, Oliver Steimer, Steffen Thier, Wolfgang Zinser, Leopold Henssler, Volker Alt, Stefan Fickert

**Affiliations:** 1Department of Trauma Surgery, University Regensburg Medical Centre, 93053 Regensburg, Germany; 2Sporthopaedicum Straubing, 94315 Straubing, Germany; 3Clinic of Orthopedics and Sports Orthopedics, Klinikum Rechts der Isar, Technical University of Munich, 81675 Munich, Germany; 4Orthopaedics, Trauma and Plastic Surgery, University Hospital Carl Gustav Carus, 01307 Dresden, Germany; 5ARCUS Sportklinik, 75179 Pforzheim, Germany; 6Clinic of Orthopedics, Saarland University Medical Center, 66421 Homburg, Germany; 7Sportchirurgie Heidelberg, ATOS Clinic Heidelberg, 69115 Heidelberg, Germany; 8Department of Orthopaedic Surgery and Traumatology, St. Vinzenz Hospital, 46535 Dinslaken, Germany; 9Department of Orthopedic Surgery and Traumatology, Medical Faculty Mannheim, University Medical Centre Mannheim, University of Heidelberg, 68167 Mannheim, Germany

**Keywords:** hip arthroscopy, microfracture, cartilage therapy, hip preserving surgery

## Abstract

Objective: According to current recommendations, large cartilage defects of the hip over 2 cm^2^ are suggested to undergo autologous chondrocyte transplantation (ACT), while small defects should be treated with microfracture. We investigated if patients with small chondral defects of the hip joint (≤100 mm^2^) actually benefit from microfracture. Design: In this retrospective multicenter cohort study 40 patients with focal acetabular cartilage defects smaller than 100 mm^2^ and of ICRS grade ≥2 caused by femoroacetabular impingement were included. Twenty-six unrandomized patients underwent microfracture besides treatment of the underlying pathology; in 14 patients cartilage lesions were left untreated during arthroscopy. Over a mean follow-up of 28.8 months patient-reported outcome was determined using the iHOT33 (international hip outcome tool) and the VAS (visual analog scale) for pain. Results: The untreated group showed a statistically significant improvement of the iHOT33 after 12 (*p* = 0.005), 24 (*p* = 0.019), and 36 months (*p* = 0.002) compared to the preoperative score, whereas iHOT33 in the microfracture group did not reveal statistically significant changes over time. There was no significant difference between both groups on any time point. Regarding pain both groups did not show a significant improvement over time in the VAS. Conclusion: The subjective outcome of patients with small cartilage defects of the hip (≤100 mm^2^) improves 12 months after arthroscopic FAIS surgery without any cartilage treatment. However, no improvement could be seen after microfracture. Therefore, a reserved surgical treatment for small cartilage defects of the hip under preservation of the subchondral bone is recommended especially if a simultaneous impingement correction is performed.

## 1. Introduction

Articular cartilage defects are a common orthopedic issue. Besides specific entities such as trauma and mechanical overload, femoroacetabular impingement syndrome (FAIS) is a major cause of hip cartilage defects and can subsequently lead to the development of hip osteoarthritis [1,2]. As the intrinsic self-regenerative capacity of hyaline cartilage is limited due to its low metabolic and proliferative activity, there is a rising demand for effective and sustainable therapeutic options in articular cartilage treatment.

The therapeutic options for cartilage defects in the hip have mainly been adapted from the knee joint and include debridement, bone marrow stimulating techniques with or without biomaterials and autologous chondrocyte transplantation, matrix-associated (MACT) or matrix-free (ACT), and two-stage (established) or one-stage (in clinical proof) [3,4]. However, concrete guidelines for cartilage therapy in the hip joint still have to be defined and proofed in hip-specific studies. Defect size may play a role in formulating joint- and technique-specific indications for the management of chondral defects in the hip. While focal and contained full-thickness lesions measuring more than 200–400 mm^2^ in size are recommended to be treated with ACT, smaller chondral defects are supposed to undergo microfracture regarding current recommendations [5,6]. However, in recent studies doubts arise if patients with small chondral defects in the hip in fact benefit from microfracture [7].

Several studies on the knee joint showed a setback of clinical outcome combined with signs of osteoarthritis two to three years after microfracture was performed, following an initial period of improving clinical results [8,9]. The deterioration of the results after microfracture is often due to the hyperossification of the subchondral border lamella, which in the worst case can lead to intralesional osteophytes. Therefore, a primary treatment with microfracture increases the failure rate of following therapy options such as ACT [10,11].

Due to the shorter history of microfracture in the hip there are only a few studies and short-term follow-up results for this joint [6,12,13,14]. However, recently Hevesi et al. compared simple debridement and microfracture in patients with focal chondral defects of the hip and reported equal results for both options regarding patient reported outcome (PRO) and revision rates at a follow-up of five years [13]. This supports the idea of a careful use of microfracture in the hip and the preference of more conservative therapy options.

In this study we investigated the clinical outcome after microfracture in the hip joint with special focus on patients with rather small acetabular cartilage defects (≤100 mm^2^) and compared them with a cohort of patients with focal acetabular cartilage lesions, which were left completely untreated during arthroscopy.

## 2. Methods

This study was designed as a retrospective register study as part of the German Cartilage Registry (KnorpelRegister DGOU), approved by the institutional review board of the University of Freiburg (No. 520/14) and the participating centers itself. Forty Patients were enrolled between January 2015 and July 2018 in 5 centers. The German Cartilage Registry is funded by the German Orthopedic and Trauma Society (DKOU), the Arthrosehilfe e.V. und Oscar-Helene-Stiftung e.V.

### 2.1. Protocol Design and Patient Cohort

The intermediate-term clinical data of patients with a focal acetabular cartilage defect of the hip in the setting of FAIS, who underwent hip arthroscopy and received either microfracture or no treatment of the chondral lesion, were assessed. The choice of treatment was unrandomized. Inclusion criteria were an age of minimum 18 years, acetabular cartilage defects ICRS grade 2 or higher caused by FAI with intact surrounding cartilage and subchondral bone, and a defect size of ≤100 mm^2^. Patients with radiographic signs of osteoarthritis higher than grade 1 according to Kellgren and Lawrence or femoral cartilage defects were excluded. Pre-operative diagnostics included a clinical examination, standardized radiographs in supine anterior–posterior view and Lauenstein projection, as well as magnetic resonance imaging with radial reconstructions. Patient relevant outcome was assessed by VAS for pain and iHOT33 (evaluation of pain and functional parameters of daily life and sports). To further evaluate symptom specific well beings, patient satisfaction and pain we added the questions listed in Table 1 to the questionnaire. The questionnaires were obtained from patients on the day before index arthroscopy (pre-operative) and at 6, 12, 24, and 36 months after arthroscopy.

### 2.2. Surgical Technique

In supine position on a traction table with about 6–8 mm joint distraction (controlled under fluoroscopy), the defect area with the localized cartilage defect was investigated utilizing two arthroscopic portals (anterolateral and anterior) and classified according to the International Cartilage Repair Society Score (ICRS). Lesion size was determined intraoperatively using a reference tool. Cartilage defects were either treated by microfracture or left completely untreated (no debridement/chondroplasty). During the microfracture procedure the defect was debrided by shaver and curettage creating a stable cartilage border and holes were made 3 to 4 mm apart and approximately 4 mm deep into the subchondral bone using an awl to access bone marrow level and achieve appropriate bleeding [15]. Concomitant corrective surgeries such as contouring of the head–neck offset, labral repair, and acetabular trimming (if necessary) were performed during the same intervention (Table 2).

### 2.3. Rehabilitation Protocol

The post-operative rehabilitation protocol mainly depended on the performed concomitant surgeries. Patients without cartilage therapy kept partial weightbearing (15 kg) for 2–4 weeks on crutches, whereas patients undergoing microfracture were prescribed crutches and partial weightbearing (15 kg) for 6–8 weeks. Afterwards patients were allowed pain-adapted full weightbearing. Patients with labral repair were restricted to a maximal flexion of 90° for 6 weeks. Continuous passive motion (CPM) therapy was conducted for 4 weeks from the first postoperative day with a minimum usage of 6 h daily. A return to competitive sport was allowed 9–12 months after operation. Furthermore, aftercare involved prophylaxis of heterotopic ossification with oral non-steroidal anti-inflammatory drug (NSAID) medication (3 × 400 mg ibuprofen daily for a period of 2 weeks) and deep vein thrombosis (DVT) prophylaxis by means of subcutaneous administration of a low-molecular-weight heparin analogue up to full weight-bearing.

### 2.4. Psychometric Analysis

Minimum clinical important difference (MCID), patient acceptable symptom state (PASS) and substantial clinical benefit (SCB) were calculated using distribution-based and anchor-based methods, respectively. One-half the standard deviation (SD) of the change in 36-month iHOT-33 scores was used to calculate the MCID. PASS and SCB values were determined by an anchor-based method using a receiver operator characteristic (ROC) analysis based on the questions in Table 1. The area under the curve (AUC) was calculated at a 95% confidence interval (CI). An AUC value greater than 0.8 and a 95% CI that does not contain 0.5 are considered excellent properties of responsiveness. The Youden index was used to optimize sensitivity and specificity values and identify the best cutoff values for the SCB change value and the PASS.

### 2.5. Statistical Analysis

Data are expressed as mean value ± standard deviation (SD). SPSS software (IBM, Armonk, NY, USA) was used for statistical analyses. Score changes between the different observation time points and treatment groups were performed on quantified data using two-sided paired *t*-tests and one-way ANOVA with Bonferroni post hoc test. A multivariate linear regression model was applied to analyze parameters influencing the change in iHOT33 and VAS. *p* < 0.05 was considered significant.

## 3. Results

A total of 40 patients (23 men/17 women) aged between 20 and 61 years (mean age 36.2 years) were included in this study. The average time of follow-up was 28.8 months (range 12–36 months). All patients were diagnosed with chondral defects of the hip joint smaller than 100 mm^2^. The mean defect size was 38.1 mm^2^. Twenty-six patients were treated with microfracture. Cartilage defects in 14 patients were left untreated. Figure 1 shows exemplary cartilage defects of each group.

The main underlying pathology, namely femoroacetabular impingement, was treated during the same intervention. CAM deformities underwent a restoring to a normal femoral head–neck offset (*n* = 38), while PINCER deformities were treated with acetabular trimming and following labrum refixation if necessary (*n* = 5). Three out of eight acetabular deformities were left untreated due to intraoperatively minor appearance. Labrum lesions were either treated with resection (*n* = 9), refixation (*n* = 20) or left untreated depending on their location and extent. The detailed demographic data and baseline characteristics of the treatment groups are illustrated in Table 2.

The preoperative investigation revealed a baseline total iHOT33 score of 41.6 ± 17.6 in the whole cohort, 42.0 ± 19.9 in the microfracture group and 42.9 ± 19.1 in the untreated group, respectively. There was no significant difference in the preoperative iHOT33 score between the two test groups.

There were no significant changes of the iHOT33 over time in the microfracture group, besides a significant improvement in the category “sport” from the 6-month control on (*p* = 0.011).

The overall iHOT33 score change in the non-treated group rises significantly after 12 (33.2 ± 27.8; *p* = 0.005), 24 months (30.8 ± 26.5, *p* = 0.019), and 36 months (33.6 ± 29.1, *p* = 0.002) related to the preoperative iHOT33 score (Figure 2). Regarding the iHOT33 subcategories, there were no significant improvements over time seen in the subdomain “job”. Categories “social” and “symptoms” of the iHOT33 increased significantly after 12, 24, and 36 months (*p* ≤ 0.030) and “sport activities” was significantly higher at the 12- and 36-month follow-up compared to the preoperative values (*p* ≤ 0.034). In the multivariate analysis the change in iHOT33 and VAS was not significantly influenced by the analyzed parameters including treatment, defect size and ICRS grade, age, BMI, sex, symptoms duration, or previous surgery.

The adjusted response rates of iHOT 33 at months 6, 12, 24, and 36 were 75.0%, 82.5%, 65.0%, and 60.0%, respectively.

The calculation of the clinically important outcome values (CIOV) for the iHOT33 at the 36-month follow-up revealed a MCID of 15, a PASS of 64, and a SCB change value of 24, which is in concordance with the current literature concerning CIOVs after hip arthroscopy [16]. The iHOT33 total score of the untreated group after 36 months was statistically significantly higher than the calculated PASS (*p* = 0.040), whereas the microfracture group did not show a significant difference (*p* = 0.894) (Figure 3).

There was an apparent reduction in the VAS score in both, the microfracture and the untreated group, from the first follow-up on. However, the improvement over time did not reach statistical significance.

The comparison of the two therapy groups showed neither significant differences of iHOT33 nor VAS for pain at any time point (Figure 4).

The overall satisfaction of all patients after surgery was constantly on a high level from the first follow-up on. While the satisfaction of the patients in the non-treated group was slightly increasing and the patients after microfracture had the tendency to become less satisfied over time, there was no significant difference between the two groups or any timepoint. During follow-up one patient of the microfracture group underwent arthroplasty of the affected hip 24 months after arthroscopy. One patient in the untreated group underwent a second arthroscopy of the affected hip for chondroplasty of the cartilage defect 36 months after index arthroscopy. There were no further treatment-related complications reported during the observation period.

Figure 2, Figure 3 and Figure 4 give a detailed overview of the iHOT33 and VAS results.

## 4. Discussion

The purpose of this study was to compare microfracture to no treatment in patients with small focal chondral defects in combination with arthroscopical FAIS surgery and/or labrum repair and to determine whether the choice of cartilage therapy was predictive of outcomes and patient satisfaction. The most important finding was that patients undergoing microfracture of their chondral defects did not significantly benefit from arthroscopy. The second group without cartilage treatment, however, showed significant improvements in symptoms and functions.

Twelve, twenty-four and thirty-six months after hip arthroscopy, the total iHOT33 and the subcategories “social”, “sport”, and “symptoms” in the non-treated group improved significantly from baseline and clinically relevant improvements were observed at each assessment timepoint, where the main improvement was achieved by month 12 and maintained afterwards, or even slightly continued to increase through month 36. The only significant improvement in the microfracture group was seen in the iHOT33 subcategory “sport”. The other subcategories as well as the total score did not change significantly over time after microfracture. Concerning the CIOVs, the change in iHOT33 score 36 months after arthroscopy in the untreated group surpassed the SCB change, while patients after microfracture hardly reached the PASS. These results may suggest that it might be better to refuse the therapy of small chondral defects of the hip joint at all before using microfracture in these cases. However, the absolute score values were not significantly different in the two groups, leading to the conclusion that a “no-therapy” strategy is at least an equal option compared to microfracture in cartilage lesions of the hip smaller than 100 mm^2^ as microfracture did not lead to a better clinical outcome. Given clinically similar findings, further treatment-related factors should be considered. As a much more invasive intervention microfracture comes along with certain disadvantages. The postoperative rehabilitation is complex compared to hip arthroscopy without cartilage therapy including a longer period of partial weightbearing and associated risks such as muscle atrophy and thrombosis. Furthermore, the perforation of the subchondral lamina of the acetabulum leads to a disruption of homeostasis of the subchondral bone and as a reaction to the high mechanical load of the hip to a higher risk of hyperossification and in the worst case to intralesional osteophyte formation, which reduces the prognosis of secondary cartilage therapies. The principle of microfracture described by Steadman et al. in 1997 [17] for the knee joint is based on the theory that blood flows out of the bone marrow through precisely performed microholes within the debrided chondral defect and forms a blood clot including mesenchymal progenitor/stromal cells (MSC) as an origin for newly formed cartilage. However, in degenerative chondral lesions Steadman generally combined the microfracture with an opening of the infrapatellar fat pad, which holds MSCs as well, to reduce scarring and improve the joint kinematics [18]. Thus, this additional procedure already guarantees the presence of MSCs in the joint and questions arise as to whether Steadman’s results can really be traced back to the microfracture in the defined defect. This effect is consciously used in primary anterior crucial ligament (ACL) repair where microfracturing of the notch area causes an intraarticular MSC release and improves healing of the sutured ACL. This technique named healing response procedure produces satisfying results in acute ACL ruptures [19]. Transferred to the hip joint, the healing response is represented by the correction of impingement deformities, which is followed by a relevant blood and MSC flow into the joint. This theory indicates that the crucial factor behind microfracture might be the surgical opening of MSC-rich tissue independently in which intraarticular location and that microfracture of the defect itself located in the weight-baring area of the joint might even have negative consequences including prolonged rehabilitation and risk of intralesional osteophytes and fractures.

Finally, due to the high-angle of the acetabulum with its deep cup-shaped structure, performing proper microfracture rather than accidentally generating uncontrolled channels can be technically demanding.

Our results are reflected within the current literature. In a matched-cohort study including 127 patients, Domb et al. compared the outcome of patients with high-grade cartilage defects (ICRS grade IV) in the hip undergoing microfracture and patients with lower-grade lesions (ICRS grade < IV) not treated with microfracture [20]. Both groups showed an overall significant improvement of the patient reported outcome after a follow-up of two years. However, there were no differences between the two groups besides the VAS score, which was significantly better in the group without microfracture treatment.

Hevesi et al. recently compared microfracture to simple debridement of high-grade acetabular cartilage lesions in 110 patients and found similar outcome scores and revision rates in the two groups [13]. Based on their results they suggest cautious treatment of small cartilage defects in the hip with the preference of debridement or abrasion over microfracture to improve rehabilitation while maintaining established favorable outcomes after hip arthroscopy.

In the microfracture group only the subdomain sport of the entire iHOT33 showed a significant improvement. An improvement in sports-related outcome categories after hip microfracture is coherent with the current literature. Several studies show high return-to-sport rates in elite athletes as well as recreational sportspersons [21,22]. However, on the one hand, in these studies control groups without microfracture achieved an equal rate of return to sport and on the other hand there is a lack of long-term results. As already reported for the knee joint [23], microfracture in the hip might have promising short-term results, in some cases facilitating a fast return to sport; however, improvements might not be sustainable and be followed by proceeding cartilage deterioration and an early ending sports career.

The following limitations existed in the study design: First, the number of patients investigated in our protocol (*n* = 40) was rather small. However, the quality of the performed procedures was homogenously high as all surgeons represent high volume hip preservation reference centers. Second, the cartilage defects in the microfracture group tended to be of higher grade, with the majority of defects having an ICRS grade III, compared to the non-treated group, which held mainly grade II lesions. This fact might reduce the comparability of the two cohorts. However, the multivariate analysis did not show a significant influence of the preoperative ICRS grade on the clinical outcome after hip arthroscopy.

Further, the choice of cartilage treatment was unrandomized and unblinded.

The defect size of <100 mm^2^ might appear rather small, especially compared with commonly treated cartilage defects in the knee joint. However, with a total joint surface of 35–55 cm^2^ [24,25] and a weight bearing area of less than 15 cm^2^, the hip joint offers much less chondral substance than the knee joint with a cartilage surface of 90–120 cm^2^ [26,27]. Therefore, even chondral defects <100 mm^2^ are highly relevant in the hip as origin of expanding joint degeneration and should be in the focus of modern cartilage therapy.

Our interpretations are restricted to clinical patient-reported outcome measurements. Follow-up arthroscopy or MRI would give insights into the regenerative potential of the examined cartilage defects; however, they are elaborate follow-up devices.

Finally, in all our patients, we performed concomitant corrective surgeries to the cartilage repair procedure to eliminate the causative problem (e.g., labral refixation, femoroplasty, acetabuloplasty). Thus, it cannot be distinguished how much of the clinical improvement is due to cartilage therapy and how much to the concomitant corrective surgery. However, the underlying pathology in most of the patients was associated with a CAM deformity, and an additional labrum lesion was present in nearly all cases. Thus, the therapy regime was the same in the majority of the patients and results are therefore comparable.

An established alternative for a simple microfracture in the hip is autologous matrix-induced chondrogenesis (AMIC) [28], which combines microfracture with subsequent implantation of a collagen membrane into the chondral defect to avoid diffusion of progenitor cells into the joint and protect the site from mechanical stress. Girolamo et al. demonstrated a significant advantage of AMIC over microfracture in treatment of cartilage lesions measuring 2–8 cm^2^ concerning long-time clinical results [29]. A recent, interesting approach even combines AMIC with a biologically active scaffold containing allograft cartilage [30]. Future studies should investigate these therapy options for chondral defects <2 cm^2^ in comparison to microfracture and simple debridement or without cartilage therapy.

## 5. Conclusions

In conclusion, our findings support that patients with small chondral defects of the hip under 100 mm^2^ undergoing hip arthroscopy achieve similar patient-reported outcome scores and satisfaction regardless of whether they underwent microfracture or no cartilage therapy. Considering the disadvantage of microfracture concerning rehabilitation and negative consequences for the subchondral bone, these outcomes support the recommendation of restricted surgical treatment of especially small chondral defects in the hip joint to optimize outcome after hip arthroscopy. In particular, if concomitant surgeries such as impingement correction affect appropriate intraarticular bleeding, the integrity of the subchondral bone within the cartilage defect should be preserved.

## Figures and Tables

**Figure 1 jcm-11-06283-f001:**
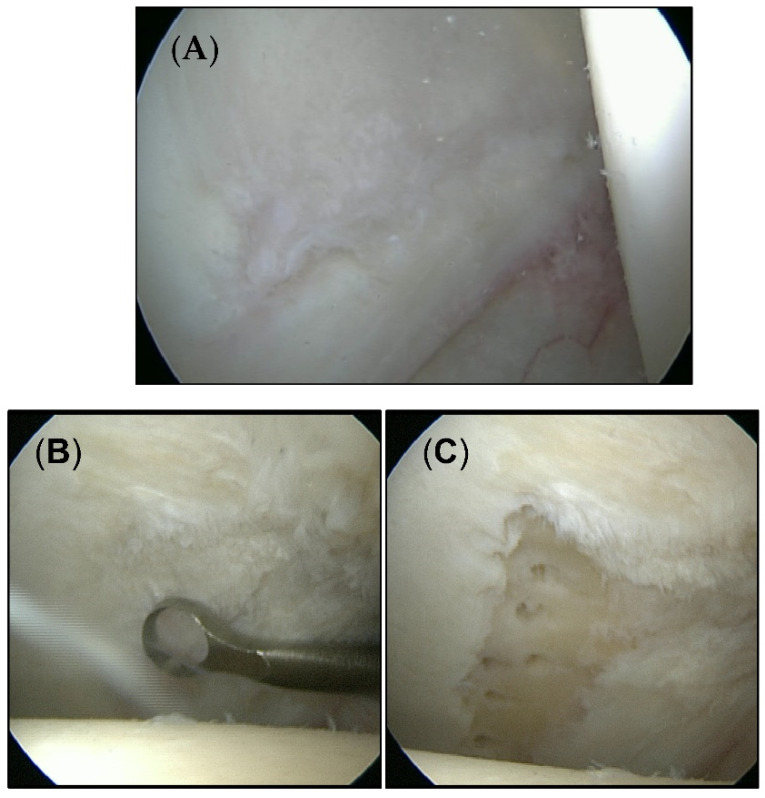
Intraoperative appearance of cartilage defects. (**A**) Untouched defects form the untreated group. (**B**) Cartilage defect from the microfracture group before (**C**) and after treatment.

**Figure 2 jcm-11-06283-f002:**
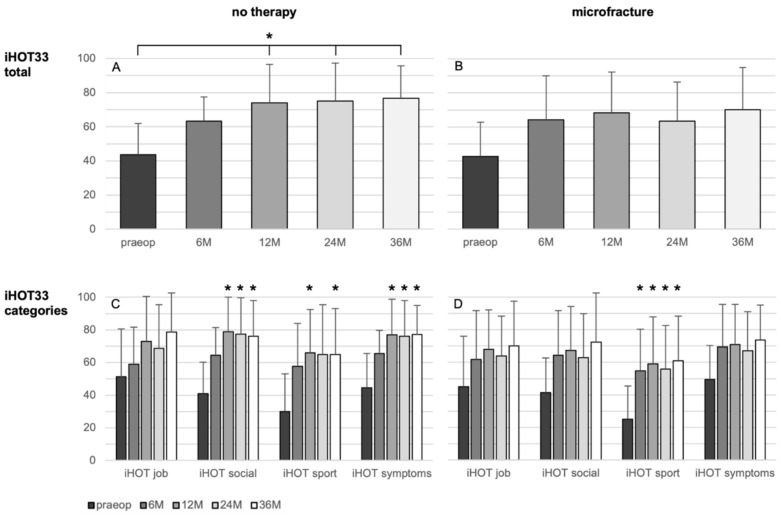
Results of the patient-reported iHOT33 score (**A**,**B**) and its subcategories (**C**,**D**) in patients undergoing microfracture (**B**,**D**) or no therapy of their hip cartilage defects during arthroscopy (**A**,**C**) from baseline to 36 months follow-up. Whiskers represent standard deviations. Significant differences (* *p* < 0.05) are designated by asterisks.

**Figure 3 jcm-11-06283-f003:**
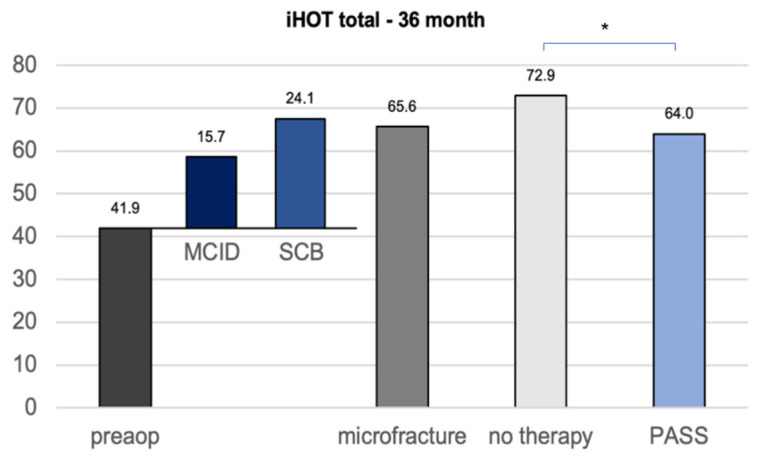
CIOVs of the iHOT33 36 months after hip arthroscopy with or without microfracture of the apparent cartilage defects. **MCID** minimum clinically important difference **SCB** substantial clinical benefit (change) **PASS** patient acceptable symptom state. Significant differences (* *p* < 0.05) are designated by asterisks.

**Figure 4 jcm-11-06283-f004:**
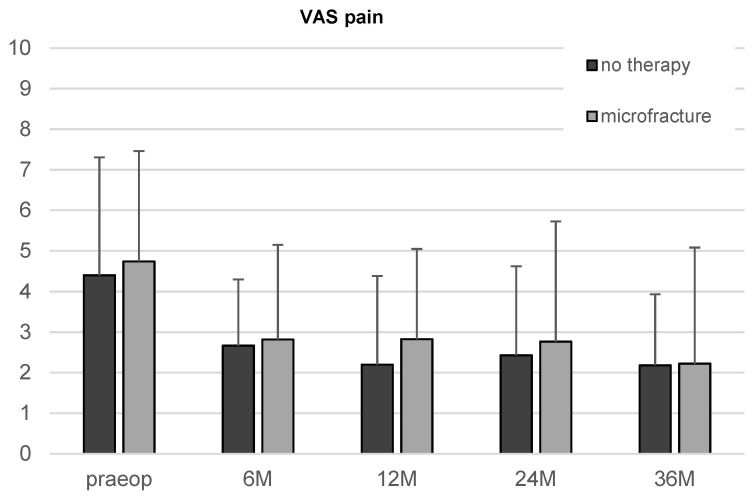
Results of the VAS for pain in patients undergoing microfracture or no therapy of their hip cartilage defects during arthroscopy from baseline to 36-month follow-up. Whiskers represent standard deviations.

**Table 1 jcm-11-06283-t001:** Questions supplementing the PRO scores regarding patient satisfaction and pain.

Question	Scoring
Are you satisfied with the result of the surgery?	1–5/not satisfied–very satisfied
How much did you benefit from the treatment?	1–5/impairing–very beneficial
How would you feel living with the current hip symptoms for the rest of your live?	1–5/not satisfied–very satisfied
Do you currently have pain in the operated hip?	1–4/no pain–strong pain
How often do you currently take pain killers?	1–5/never–daily

**Table 2 jcm-11-06283-t002:** Demographic data and baseline characteristics of the study population (*n* = 40).

		No Therapy (*n* = 14)	Microfracture (*n* = 26)	Total (*n* = 40)
Sex				
Female	*n* (%)	7 (50.0)	10 (38.5)	17 (42.5)
Male	*n* (%)	7 (50.0)	16 (61.5)	23 (57.5)
Age (years)	Mean ± SD	36.2 ± 13.8	36.1 ± 9.1	36.2 ± 10.7
BMI (kg/m^2^)	Mean ± SD	23.1 ± 2.6	25.1 ± 3.8	24.4 ± 3.1
Smoking status				
Yes	*n* (%)	3 (21.4)	7 (26.9)	10 (25.0)
No	*n* (%)	11 (78.6)	19 (73.1)	30 (75.0)
Symptoms duration (months)	Mean ± SD	27.6 ± 16.0	24.8 ± 25.0	25.8 ± 21.8
Follow-up (months)	Mean ± SD	33.4 ± 6.7	26.3 ± 10.0	28.8 ± 9.6
Defect size (mm^2^)	Mean ± SD	33.7 ± 15.0	39.9 ± 23.6	38.1 ± 21.1
Defect number				
1	*n* (%)	11 (78.6)	23 (88.5)	34 (85.0)
2	*n* (%)	2 (14.3)	3 (11.5)	5 (12.5)
3	*n* (%)	1 (7.1)		1 (2.5)
ICRS grading				
Grade II	*n* (%)	12 (85.7)	4 (15.4)	16 (40.0)
Grade III	*n* (%)	2 (14.3)	22 (84.6)	24 (60.0)
Underlying pathology				
CAM deformity	*n* (%)	6 (42.9)	18 (69.2)	24 (60.0)
CAM deformity + dysplasia	*n* (%)	5 (35.7)	3 (11.5)	8 (20.0)
PINCER deformity	*n* (%)	1 (7.1)	1 (3.8)	2 (5.0)
Combined FAIS	*n* (%)	2 (14.3)	4 (15.4)	6 (15.0)
Labrum lesion				
Yes	*n* (%)	13 (92.9)	26 (100.0)	39 (97.5)
No	*n* (%)	1 (7.1)		1 (2.5)
Concomitant surgeries				
Labrum resection	*n* (%)	4 (28.6)	5 (19.2)	9 (22.5)
Labrum refixation	*n* (%)	8 (57.1)	12 (46.2)	20 (50.0)
Labrum trimming	*n* (%)	1 (7.1)	9 (34.6)	10 (25.0)
Acetabular trimming	*n* (%)	3 (21.4)	2 (7.7)	5 (12.5)
Femoral neck contouring	*n* (%)	13 (92.9)	25 (96.2)	38 (95.0)
Previous surgery of affected hip				
Yes	*n* (%)	2 (14.3)	2 (7.7)	4 (10.0)
No	*n* (%)	12 (85.7)	24 (92.3)	36 (30.0)

## Data Availability

The data underlying this article will be shared on reasonable request to the corresponding author.

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
