# Peer review of "Patients with Small Acetabular Cartilage Defects Caused by Femoroacetabular Impingement Do Not Benefit from Microfracture"

_jcm, 2022, doi:10.3390/jcm11216283_

Round 1
Reviewer 1 Report
Thank you very much for your very clinically valuable study and for submitting it to this journal.
Why do you think the microfracture group showed significant improvement (p=0.011) in the "sports" item at the postoperative 6-month? If you could add that idea to the discussion, etc., it would help us understand better.
Shouldn't it be mentioned that this study was not randomized as a limitation of this study? Who made the decision to microfracture or not and how? And was this reported to the patients? If so, did it affect the patient-oriented outcome?
You mention the possibility that the microfracture may have had a negative effect, but is it possible that the hip arthroscopy itself was not very effective and that conservative treatment would have produced similar results? Shouldn't there be a No surgery group the next time a similar study is conducted?
It would be a better paper if you could describe above things. Thank you in advance.
Reviewer 2 Report
Comments to the Author
I applaud the authors for a solid study and well-written and constructed manuscript. My main concern is that this study does not significantly contribute to or alter our daily practice and our knowledge of hip arthroscopy. As described in the manuscript, previously published data (references 11, 12, 16, 22) already reported that patients undergoing debridement/abrasion of acetabular cartilage lesions demonstrate similar outcome scores and revision rates compared with those of patients undergoing microfracture. Moreover, some of these studies were multicentered, reporting on more than a hundred patients with a higher average follow-up period (4.9 years Vs 28.8 months in the current study). Lastly, it was a retrospective and unrandomized study; the authors do not disclose the methods they used to divide the two groups, which may raise the suspicion of selection bias.
In summary, it is challenging to see the contribution of this study to the existing literature.
Abstract
Well written and coherent.
Introduction
Lines 46-49: The sentence needs to be rewritten because it's coherent enough.
Lines 59-60: Please see the opening comment
The authors cite two papers (references 11 and 12), the first is a review paper, and the second is a more recent multi-center study reporting on 113 hips with a mean follow-up of 4.9 years. As cited in the current study, both papers reported that patients undergoing debridement/abrasion of acetabular cartilage lesions demonstrate similar outcome scores and revision rates compared with those of patients undergoing microfracture. This study was carried out in a single center, involving 40 hips with a 28.8 months follow-up. What is the contribution of this study to the existing literature?
Methods
General:
1) I could not find how the authors measured the size of the chondral defect?, and from which portals.
2) How many surgeons performed the surgery?
Protocol design and patient cohort
Line 85: It was a retrospective and unrandomized study. May the authors elaborate more on the process?. For example, how were the patients allocated to each group?, was it before surgery?, and if so, what were the criteria?.
Results
Line 139: ‘The average time of follow-up was 28.8 months’; what was the minimum follow-up?
Lines 141-142: As previously discussed, why 26 patients versus 14?, what were the criteria for this division?
Line 156: Table 2, can the authors provide the location of the chondral damage (clockface or another method) and its proximity to the chondral labral junction?
Discussion
Lines 283-289: Well-constructed and persuasive paragraph
Limitations
Please add a limitation section
References
Please add more up-to-date references such as:
PMID: 31069090 (DOI: 10.1093/jhps/hnz002) or PMID: 29868405 (PMC5982236)
Reviewer 3 Report
Dear Editor, Dear Authors,
Thank you for giving me the opportunity to review this interesting article.
Unfortunately, from my point of view, the draft cannot be published in this form and needs major revisions.
Title
Please do not formulate the title as a question, be proactive here. Perhaps acetabular would be the more accurate wording than hip (alone).
Abstract
Please check if the statements in the abstract are consistent with the article. Please formulate the abstract in passive voice.
Text
Please formulate in passive voice.
Introduction
Are there differences between traumatic cartilage defects and degenerative cartilage defects? Please add if necessary.
L43: Paragraph Control
Methods
L 77: The ethics approval states a retrospective study. However, L 95 explains a prospective design. Can you explain this in more detail?
L79f: The time period is 2015-2018, please provide when the last inclusion and when the last 36m survey was done in the methods section.
L87: How and when was the defect size determined that led to inclusion? MRI? Intraoperative? How and when was the preoperative survey performed?
What was done with patients with femoral cartilage defects?
L99: How was the traction controlled? Estimated under flouroscopy, or measured? Were there force gauges controlling the traction?
L102: You state your measurements to one decimal (L141). Please describe the measuring device in more detail, especially how accurately this measuring device can be used to measure.
L103: Please describe the procedure in more detail. Untreated really means untreated? Were cartilage margins debried? What happened to loose portions? The cartilage base? How was microfracturing performed (reference if applicable)?
L109: Please explain the post-treatment scheme in more detail for the two types of treatment. On crutches, different weight-bearing, unweighting, pain-adapted full weight-bearing can be applied.
Results
The results are presented conclusively.
Discussion
The discussion is compliant.
Overall, points remain open. Ultimately, the procedure was worthwhile for patients in both treatment arms, leading to pain reduction and a better score postoperatively. It should be more stringently elaborated that in small defects, microfracturing does not lead to a better clinical outcome.
Information as to whether the procedure has an influence on the cartilage substance of the patient or possibly protects against endoprosthesis remains open. This could be mentioned in the limitations. Another limitation would be that no distinction was made between fresh traumatic and older degenerative damage.
References
The literature given is not very current. Please check for more recent references.
References 26-29 are missing. Please Check
Round 2
Reviewer 2 Report
Thank you for your detailed reply; after reading it, I have decided to recommend accepting the paper.
Reviewer 3 Report
The revisions and additions made by the authors are sufficent and have rounded out the article well. I would like to recommend the article for publication.